# Peephole Steering: Speed Limitation Models for Steering Performance in Restricted View Sizes

Shota Yamanaka*
Yahoo Japan Corporation

Hiroki Usuba
Meiji University

Haruki Takahashi
Meiji University

Homei Miyashita
Meiji University

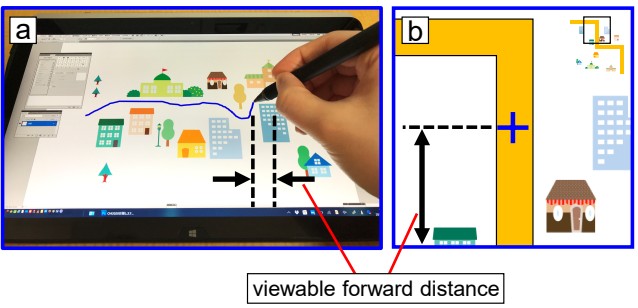

Figure 1: Examples of steering through narrow paths with limited forward views. (a) Lasso operation for selecting multiple objects in illustration software. The user's hand occludes the forward path to be passed through. The viewable forward distance is between the stylus tip and the user's hand. (b) Map navigation in a zoomed-in view proposed in a previous study [18]. When the cursor moves downwards, the viewable forward distance is between the cursor and the window bottom.

## ABSTRACT

The steering law is a model for predicting the time and speed for passing through a constrained path. When people can view only a limited range of the path forward, they limit their speed in preparation of possibly needing to turn at a corner. However, few studies have focused on how limited views affect steering performance, and no quantitative models have been established. The results of a mouse steering study showed that speed was linearly limited by the path width and was limited by the square root of the viewable forward distance. While a baseline model showed an adjusted $R^2 = 0.144$ for predicting the speed, our best-fit model showed an adjusted $R^2 = 0.975$ with only one additional coefficient, demonstrating a comparatively high prediction accuracy for given viewable forward distances.

**Index Terms:** H.5.2 [User Interfaces]: User Interfaces—Graphical user interfaces (GUI); H.5.m [Information Interfaces and Presentation]: Miscellaneous

## 1 INTRODUCTION

The steering law [1, 14, 27] is a model for predicting the time and speed needed to pass through a constrained path, such as navigation through a hierarchical menu. In HCI, the validity of the steering law has typically been confirmed in a desktop environment, such as by maneuvering a mouse cursor or stylus tip through a path drawn on a display (e.g., [2, 31]). Under such conditions, participants can view the entire path or a substantial portion of it before the trial begins, and they can thus determine the appropriate movement speed for a

---

*e-mail: syamanak@yahoo-corp.jp

given path width.

However, conditions under which users can view enough of a long path forward represent an ideal situation. Imagine a user operating a stylus pen in illustration software to select multiple objects with a lasso tool as shown in Fig. 1a. When a right-handed user moves the stylus rightwards, the *viewable forward distance* is limited due to occlusion by the user's hand. Therefore, to avoid selecting unwanted objects, the user has to move the stylus slowly. In contrast, if the user moves the stylus leftwards through the objects, the movement speed should be less limited because the viewable forward distance is not restricted.

Therefore, for path steering tasks, we assume that the viewable forward distance limits the movement speed as the path width does. Although the effect of limited view sizes has been investigated several times in HCI studies [10, 16, 21, 30], the main interest has been target selection. Furthermore, for steering tasks, while the view of the forward path may be limited, we found few papers on this topic that include map navigation tasks with a magnified view (Fig. 1b, [18]).

If we can derive models of the relationship between task conditions and outcomes — path width and viewable forward distance vs. movement speed — it would contribute to better understanding of human motor behavior. However, evaluating the model robustness of the steering law against an additional constraint (viewable forward distance) has not been investigated well; this motivated us to conduct this work. In our user study, we conducted a path-steering experiment with a mouse and determined the best-fit model from among candidate formulations. Our key contributions are as follows.

**(a)** We provide empirical evidence that the viewable forward distance $S$ significantly affects the steering speed. We also justify why the relationship between $S$ and speed can be represented by the power law.

**(b)** We develop refined models to predict movement speed on the basis of the path width and $S$, which had significant main effects and interaction effects. Our model predicts the speed with an adjusted $R^2 > 0.97$. We also show that the movement time while steering through a view-limited area can be predicted with $R^2 > 0.97$.

We also discuss other findings, e.g., the reason a conclusion opposite of those from previous studies was obtained: the speed *increased* with a narrower $S$ in peephole pointing [21].

## 2 RELATED WORK

### 2.1 Steering Law Models

Rashevsky [27, 28], Drury [14], and Accot and Zhai [1] proposed a mathematically equivalent model to predict the movement speed when passing through a constant-width path:

$$V = \text{const} \times W \quad (1)$$

where $V$ is the speed and $W$ is the path width. Typically, participants are instructed to perform the task as quickly and accurately as possible. Hence, there are several interpretations of $V$: the possible maximum safe speed $V_{max}$ in Rashevsky's model, the average speed

$V_{avg}$ in a given path length in Drury's model (i.e., $V_{avg} = A/MT$, where the path length is $A$ and the movement time needed is $MT$), and the instantaneous speed at a given moment in Accot and Zhai's model.

The validity of this model ($V \propto W$) has been empirically confirmed for (e.g.) car driving [8, 12, 15], pen tablets [39], and mice [31, 37]. Because $V_{avg}$ is defined as $A/MT$, the following equation for predicting $MT$ is also valid [14, 20]:

$$MT = b(A/V_{avg}) \tag{2}$$

where $b$ is a constant (hereafter, $a$–$e$ indicate regression coefficients, with or without prime marks, as in $b'$). Since $V_{avg} = \text{const} \times W$, Equation 2 can be written as follows.

$$MT = b\frac{A}{\text{const} \times W} = b'\frac{A}{W} \quad (\text{let } b' = b/\text{const}) \tag{3}$$

For predicting both $V_{avg}$ and $MT$, these no-intercept forms are theoretically valid although the contribution of the intercept is often statistically significant [20] as follows.

$$V_{avg} = a + bW \tag{4}$$
$$MT = a + b(A/V_{avg}) \tag{5}$$
$$MT = a + b(A/W) \tag{6}$$

The steering law models on $V_{avg}$ and $MT$ hold when $W$ is narrow relative to the path length. Otherwise, users do not have to pay attention to path boundaries, in which case $W$ does not limit the speed [1, 20, 33]. For mouse steering tasks, $W$ limits the speed when the steering law difficulty ($A/W$) is greater than 10 [31, 33]. Hence, in our user study, we chose the range of $A/W$ ratios for the speed measurement area to include values less than and greater than 10 so that the priority for limiting the movement speed would change between $W$ and $S$. That is, if $W$ is small and $A/W$ is greater than 10, we assume that $W$ strongly limits the speed, whereas if $W$ is sufficiently large such that the path width does not restrict the speed, we assume that $S$ restricts the speed more.

## 2.2 Steering Operations with Cornering

To accurately predict the $MT$ for steering around a corner as shown in Fig. 1a and b, Pastel [26] refined the model by adding the Fitts' law difficulty [17] as follows.

$$MT = a + b\frac{2A}{W} + cID \tag{7}$$

where the first and second path segments before/after the corner have the same length ($A$) and same width ($W$), and $ID$ is the index of difficulty in Fitts' law (Pastel used the Shannon formulation [23]: $ID = \log_2(A/W + 1)$). Fitts' law was originally a model for pointing to a target with width $W$ at distance $A$. Therefore, in addition to considering the difficulty of steering in order to pass through the entire path, this model also considers the difficulty of decelerating to turn at a corner. However, if users cannot see an approaching corner due to the restricted view, it is difficult to start to decelerate with appropriate timing.

## 2.3 Effect of Viewable Forward Distance on Task Performance

Peephole pointing [10,21] and magic lens pointing [30] are examples of UI operations with restricted view sizes. The most popular task for peephole pointing is map navigation. When users want to see information about a landmark on a map application using a smartphone or PC, they first scroll the map (*search phase*) and then select an intended location (*selection phase*). Because Fitts' law [17] holds for 1D scrolling tasks done to capture a target into the viewing area [19],

the $MT$ changes due to $S$, and the total time can be predicted by the sum of the search and selection phases [10,30]. Models for peephole pointing have been validated with a mouse [10], spatially aware phone [30], handheld projector [16,21], and touchscreen [41].

Although the importance of user performance models for the peephole situation is explained in these papers, their main focus has unfortunately been on target selection. An exception that studied the effect of the viewable range in steering-law tasks was the work of Gutwin and Skopik [18], in which an area around the cursor was zoomed in on with radar-view tools (see Figure 3 in [18]). The cursor and view window were moved concurrently, and users moved the window to steer the cursor through a path.

There are two differences between the work of Gutwin and Skopik [18] and ours. First, they fixed the window size of the radar view. Thus, as the zoom level increased, the corresponding viewable forward distance $S$ decreased. In contrast, in our intended tasks (Fig. 1), $S$ changes, but there is no zooming. They reported that the zoom level did not substantially change $MT$, which is the opposite conclusion of that reached in the peephole pointing studies. In other words, a consistent effect of $S$ on $MT$ was not observed for steering and pointing; we revisit this point in the Discussion. The second difference is that the entire view was provided as a *miniature view* (see Fig. 1b), which assisted the timing for deceleration in preparation for the next corner.

In summary, the quantitative relationship between steering performance ($V_{avg}$ or $MT$) and $S$ is unclear. Yet, knowing this relationship would be beneficial for understanding user behavior in restricted-view situations, which are realistic for some tasks as described in Introduction. We tackle this challenge through a user study.

## 3 MODEL DEVELOPMENT FOR UNCERTAIN CORNERING TIMING

As the baseline model for predicting the movement speed, we test Equation 4 ($V_{avg} = a + bW$) on our experimental data. Also, to check the effect of an additional task parameter (here, $S$) on the estimated result ($V_{avg}$), the simplest method is to add the additional factor and the interaction term between the two predictor variables (if the interaction term is significant) to the baseline model[1]. Thus, we test:

$$V_{avg} = a + bW + cS \tag{8}$$

$$V_{avg} = a + bW + cS + dWS \tag{9}$$

We next discuss how users limit the speed as they prepare for a corner. As the first step towards deriving a more general model, in this study we fix the width of the second path segment $W_2$, and we give the experimental participants the previous knowledge ($PK$) of $W_2$ being fixed. Nevertheless, the participants do not know the amplitude of the first path segment, and thus the corner appears at an uncertain time.

We incorporate Pastel's model in which users must decelerate in the first path segment when approaching the corner to safely enter the second path segment [26]. As a more general case, the first and second path segments have different lengths and widths as shown in Fig. 2a. Pastel's idea for integrating Fitts' $ID$ is that the cursor must stop within the second path area, which has a width of $W_2$, after traveling over the first path segment. Hence, Equation 7 can be rewritten as:

$$MT = a + b\frac{A_1}{W_1} + cID + d\frac{A_2}{W_2} \tag{10}$$

where $ID = \log_2(A_1/W_2 + 1)$. That is, the movement amplitude for pointing is the distance of the first path and the target size is the width of the second path.

---

[1]This is explained in introductory statistics textbooks or websites, e.g., https://web.archive.org/web/20190617154140/https://www.cscu.cornell.edu/news/statnews/stnews40.pdf.

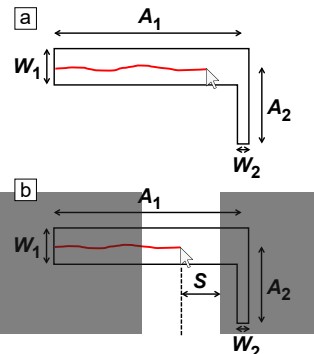

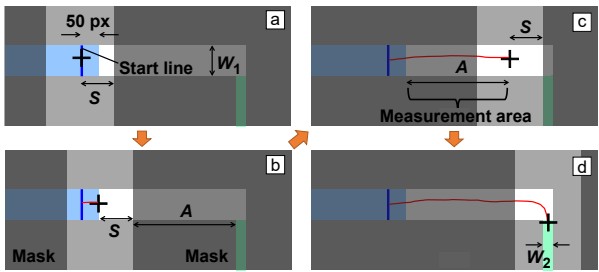

Figure 2: Steering operations with cornering in which the first and second path segments have different sizes. (a) No-mask and (b) masked conditions. Left and right masks are opaque in study, rather than semi-transparent as shown here.

Figure 3: Visual stimuli used in the experiment. Left and right masks are opaque in study, rather than semi-transparent as shown here.

As shown in Fig. 2b, when a corner has not yet been revealed from the forward mask, users can at least move over a viewable forward distance $S$. In the case that the corner is just beyond the viewable forward distance, users must adjust the speed as if the second path tolerance ranged from $S$ to $S+W_2$; the "target" center of the second path segment is located at a distance of $S+0.5W_2$ from the cursor position. The time needed to perform this pointing motion is modeled by, according to Pastel, Fitts' law with $S+0.5W_2$ as the target amplitude and $W_2$ as the width. Another definition of the amplitude in Fitts' law is the distance to the closer edge of the target, which holds empirically [3,4,6]. Using $S$ as the amplitude is thus a simpler choice that does not degrade model fitness.

If $W_2$ is sufficiently wide, it is possible for users to move the cursor rapidly in the first path segment because they can appropriately decelerate as soon as they notice the corner. However, such a task is not considered to be a steering task in a constrained path, and it is necessary that the path widths ($W_1$ and $W_2$) are not extremely wide for the steering law to hold [1, 14, 20]. In this study, we therefore set a reasonably narrow $W_2$ that necessitates careful movements to safely turn at the corner.

Another model for pointing tasks is by Meyer et al. [25].

$$MT = b\sqrt{A/W} \qquad (11)$$

where $A$ is the distance to the target center. While the mathematical equivalency between this power model and Fitts' logarithmic model is questioned by Rioul and Guiard [29], they agree that these models are well approximated.

On the basis of this discussion, we assume that in practice the $MT$ for pointing to the second path segment, which might be just beyond the front mask and which ranged from $S$ to $S+W_2$, can be regressed as follows:

$$MT = b\sqrt{S/W_2} \qquad (12)$$

Again, the original model of Meyer et al. uses the distance to the target center as the target amplitude ($S+0.5W_2$), but using $S$ as amplitude would also fit well. The average speed for this movement is defined as the distance to be traveled divided by the time needed for travel.

$$V_{avg} = \frac{S}{MT} = \frac{S}{b\sqrt{S/W_2}} = b'\sqrt{S \times W_2} \ \text{(let } b' = 1/b) \qquad (13)$$

In our experiment, to focus on the new factor $S$, we fixed the value of $W_2$. Equation 13 can thus be further simplified:

$$V_{avg} = b'\sqrt{S \times W_2} = b''\sqrt{S} \ \text{(let } b'' = b'\sqrt{W_2}) \qquad (14)$$

In summary, we hypothesize that when the viewable forward distance is limited to $S$, users have to limit the speed in case the second path segment is just beyond the viewable forward distance, and this behavior is expected to be modeled as $V_{avg} = b\sqrt{S}$. While this model is justified based on existing theoretical and empirical evidence, we need to test the validity of our hypothesis empirically. We therefore conduct a path-steering study to evaluate the model combined with the steering law.

## 4 EXPERIMENT

### 4.1 Participants

Twelve university students participated (3 females and 9 males; $M$ = 21.6, $SD$ = 1.32 years). All were right-handed and had normal or corrected-to-normal vision. Six were daily mouse users.

### 4.2 Apparatus

The PC was a Sony Vaio Z (2.1 GHz; 8-GB RAM; Windows 7). The display was manufactured by I-O DATA ($1920 \times 1080$ pixels, 527.04 mm $\times$ 296.46 mm; 60-Hz refresh rate). A Logitech optical mouse was used (model: G300r; 1000 dpi; 2.05-m cable) on a large mouse pad (42 cm $\times$ 30 cm). The experimental system was implemented with Hot Soup Processor 3.5 and was used in full-screen mode. The system read and processed input $\sim$125 times per sec.

The cursor speed was set to the default: the pointer-speed slider was set to the center in the Control Panel. Pointer acceleration, or the *Enhance Pointer Precision* setting in Windows 7, was enabled to allow mouse operations to be performed with higher ecological validity [11]. Using pointer acceleration does not violate Fitts' law or the steering law [2, 36]. The large mouse pad and long mouse cable were used to avoid *clutching* (repositioning of the mouse) during trials. This was to omit unwanted factors during model evaluation. If we had allowed clutching and the model fit was poor, we would not have been able to determine whether the poor fit was due to the model formulation or to clutching. No recognizable latency was reported by the participants.

### 4.3 Task

The participants had to click on the blue starting line, horizontally steer through a white path of width $W_1$, turn downwards at a corner, and then enter a green end area (Fig. 3). After that, an orange area labeled "Next" appeared on the left-side screen edge; entering this area started the next trial. Because the direction of cornering was not a focus of this study and because a previous study showed that the $MT$ for downward turns was shorter than that for upward turns [33], we chose downward movement for the second path segment to shorten the duration of the experiment.

If the cursor entered the gray out-of-path areas, a beep sounded, and the trial was flagged as a steering error $ER_{steer}$ and retried later. If the cursor did not deviate from the blue and white path segments, the trial was flagged as a *success*, and when entering the green end

area a bell sounded (no clicking was needed). The left and right masks moved alongside the cursor.

The participants were instructed to not make any errors and to move the cursor to the end area in as short a time as possible. In addition, we asked them to refrain from clutching while steering. If the participants accidentally clutched or if the mouse reached the right edge of the mouse pad, they were instructed to press the mouse button. Such trials were flagged as *invalid* and removed from the data analysis. If a steering error or an invalid trial was observed, a beep sounded, and the trial was presented again later in a randomized order.

The *measurement area* of distance $A$ for recording $MT$ and speed is shown in Fig. 3c; that is, it ranges from (b) when the cursor reaches the left edge of the white path to (c) when the viewable range is one pixel away from the second path segment. While in this area, the participants did not know the position of the corner and thus had to move the cursor carefully to avoid deviating from the path. We measured the $MT$ spent in the *measurement area*, and the average speed, which was the dependent variable, was computed as $V_{avg} = A/MT$. Because in the *measurement area* the participants could not see the corner, the only operation required in this area was to steer through a constrained path with a restricted view.

To avoid revealing the position of the corner before the trial began, (1) the cursor had to be moved to the "Next" area at the left edge of the screen at the end of every trial, and (2) the cursor had to stop at the blue starting line and could not move further rightwards until the line was clicked. We provided a *run-up area* of 50 pixels (Fig. 3a) because the speed when clicking on the blue starting area was zero. When the speed measurement began, the cursor was already moving at some speed.

## 4.4 Design and Procedure

This experiment had a $6 \times 5$ within-subjects design with the following independent variables and levels. We tested six $S$ values: 25, 50, 100, 200, and 400 pixels and the no-mask condition. The no-mask condition was included to measure baseline performance. The $W_1$ values were 19, 27, 37, 49, and 63 pixels. The movement time and speed were measured in the area shown in Fig. 3c. The average speed was computed as $V_{avg} = A/MT$ and used as the dependent variable.

The width of the end area $W_2$ was fixed at 19 pixels. To prevent participants from noticing that the corner appeared at several fixed positions, we used various $A$ values, and in every trial the starting line had a random offset, ranging from 100 to 400 pixels, from the left-side screen edge. The y-coordinate of the white path center had a random offset ranging from $-150$ to 150 pixels from the screen center. The $A$ values for the measurement area were 300, 500, and 800 pixels and were not included as an independent variable. For the no-mask condition (baseline), the white-area distance was set to $A + 400 + W_2$ pixels (i.e., same as the largest $S$ condition).

The ratio of $A/W_1$ ranged from 4.76 to 42.1. As discussed in Related Work, we chose the $A$ and $W_1$ values so that the $A/W_1$ ratio would range from less than to greater than 10 in order to change the motivation for limiting the speed between $W_1$ and $S$. The $W_2$ value was then set to the smallest $W_1$ value to require a careful cornering motion.

Among the combination of $6_S \times 5_{W_1} \times 3_A = 90$ patterns, 10 trials were randomly selected as practice trials. The participants then attempted three sessions of 90 data-collection trials. In total, we recorded $90_{patterns} \times 3_{sessions} \times 12_{participants} = 3240$ successful data points. This study took approximately 40 min per participant.

## 4.5 Results

For the error rate analysis, we used non-parametric ANOVA with the Aligned Rank Transform (*ART*) [35] and Tukey's method for p-value adjustment in posthoc comparisons. For the speed data analysis, we

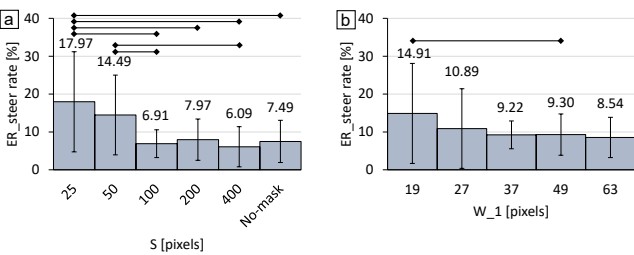

Figure 4: Results for error rate over the entire trial of the experiment.

used repeated-measures ANOVA and the Bonferroni correction as the p-value adjustment method. Note that ANOVA is robust even when experimental data are non-normal [7].

### 4.5.1 Errors

*Steering Error over the Entire Trial.* The number of $ER_{steer}$ errors for the smaller to larger $S$ values were 144, 103, 41, 49, 37, and 46, respectively. The number of $ER_{steer}$ errors for narrower to wider $W$ values were 126, 85, 70, 74, and 65, respectively. The mean $ER_{steer}$ rate was 11.5%, which was slightly higher than that found in previous work on mouse steering tasks (9% [2]). Based on the experimenter's observation, the point with the highest error rate was at the corner.

We observed the main effects of $S$ ($F_{5,55} = 7.830$, $p < 0.001$, $\eta_p^2 = 0.42$) and $W_1$ ($F_{4,44} = 3.529$, $p < 0.05$, $\eta_p^2 = 0.24$) on the $ER_{steer}$ rate in the entire trial. Fig. 4 shows these results. Post-hoc tests showed significant differences between six pairs of $S$ values: (25, 100), (25, 200), (25, 400), (25, no-mask), (50, 100), and (50, 400) with $p < 0.05$ for all pairs. $W_1 = 19$ and 49 also showed a significant difference ($p < 0.05$). No significant interaction was found between $S$ and $W_1$ ($F_{20,220} = 1.539$, $p = 0.07055$, $\eta_p^2 = 0.12$).

We expected the $ER_{steer}$ rate to decrease for greater $S$ values because the risk of deviating from the first path segment is lower in those cases. However, at the same time, the participants were able to move the cursor more rapidly as shown in Section 4.5.2 with greater $S$. Rapid movement increased the possibility of deviating from the path, and thus Fig. 4 does not show a monotonic tendency.

*Steering Error in the Measurement Area.* The number of $ER_{steer}$ errors for smaller to larger $S$ were 12, 9, 18, 15, 9, and 10, respectively. The number of $ER_{steer}$ error for narrower to wider $W$ were 40, 13, 9, 9, and 2, respectively. The mean $ER_{steer}$ rate was 2.20%. We observed the main effects of $S$ ($F_{5,55} = 15.05$, $p < 0.001$, $\eta_p^2 = 0.58$) and $W_1$ ($F_{4,44} = 9.362$, $p < 0.05$, $\eta_p^2 = 0.46$) on the $ER_{steer}$ rate in the measurement area. Fig. 5 shows these results. Post-hoc tests showed significant differences between eight pairs of $S$ values: (25, 100), (25, 400), (50, 100), (50, 200), (100, 200), (100, 400), (100, no-mask), and (200, 400) with $p < 0.05$ for all pairs. Also, four pairs showed significant differences for $W_1 = 19$ and the other values with $p < 0.05$. The interaction of $S \times W$ was significant ($F_{20,220} = 3.262$, $p < 0.001$, $\eta_p^2 = 0.23$). Fig. 6 shows this result.

### 4.5.2 Average Speed

We found the main effects of $S$ ($F_{5,55} = 69.98$, $p < 0.001$, $\eta_p^2 = 0.86$) and $W_1$ ($F_{4,44} = 180.6$, $p < 0.001$, $\eta_p^2 = 0.94$) on $V_{avg}$ to be significant. The $V_{avg}$ values for $S = 25$–400 pixels and the no-mask condition were 154, 224, 320, 420, 520, and 545 pixels/sec, respectively. Post-hoc tests showed significant differences between all $S$ pairs (at least $p < 0.01$) except for one pair ($S = 400$ and the no-mask condition). The $V_{avg}$ values for $W_1 = 19$–63 pixels were 240, 302, 365, 428, and 485 pixels/sec, respectively. Significant differences were found for all $W_1$ pairs ($p < 0.001$).

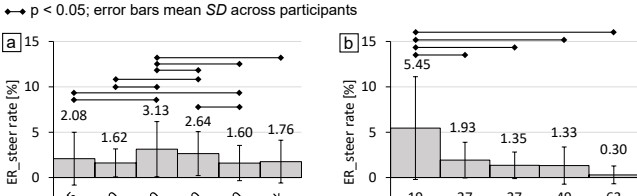

Figure 5: Results for error rate in the measurement area of the experiment.

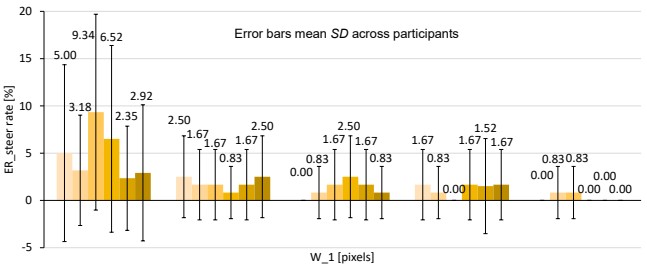

Figure 6: Interaction of error rate in the measurement area of the experiment. The six bars in each cluster show results for $S$ = 25, 50, 100, 200, and 400 pixels and the no-mask condition, respectively, from left to right.

The interaction of $S \times W_1$ was significant ($F_{20,220} = 48.70$, $p < 0.001$, $\eta_p^2 = 0.82$). As shown in Fig. 7a, we observed the following results.

- For $S$ = 200 and 400 pixels and the no-mask condition, $V_{avg}$ decreased as $W_1$ decreased. This means that **when the viewable forward distance is long, the path width can restrict the speed** following the steering law.

- However, as $S$ decreased, the effect of $W_1$ in limiting the speed tended to be smaller, i.e., the $V_{avg}$ differences became insignificant for more $W_1$ pairs. This indicates that **when the viewable forward distance is short, the speed is already limited by $S$, and thus the speed is not largely affected by changes in $W_1$**.

Furthermore, Fig. 7b shows that for all five $W_1$ values, there were no significant differences between $S$ = 400 pixels and the no-mask condition. Therefore, in our experimental setting, $S$ = 400 pixels was sufficient to eliminate the effect of the masks on $V_{avg}$. If we had included longer $A$ values, however, it is possible that $V_{avg}$ for the no-mask condition would have been much higher. It is thus fair to avoid concluding that the steering performance for $S$ = 400 pixels is equivalent to that for the no-mask condition.

To analyze the speed profiles in the measurement area, we re-sampled the cursor trajectory every 25 pixels to reduce noise in the raw data. Fig. 8a shows that speed changes for all five $W_1$ values are not evident for the narrowest $S$, similarly to Fig. 7a. Then, as $S$ increases, the effects of $W_1$ on $V_{avg}$ are exhibited more clearly (Fig. 8b and c). In the same manner, Fig. 8d–f show that the effects of $S$ become clearer as $W_1$ increases.

Fig. 9a and b show that the power model is more appropriate than the linear model for predicting $V_{avg}$ with $S$. Here, we merged the five $W_1$ values for the purposes of clearly illustrating the prediction accuracy when using $S$ and $\sqrt{S}$. Similarly, Fig. 9c shows a high correlation between $V_{avg}$ and $W_1$ in the restricted-view conditions; this plot merges the five $S$ conditions (excluding the no-mask condition).

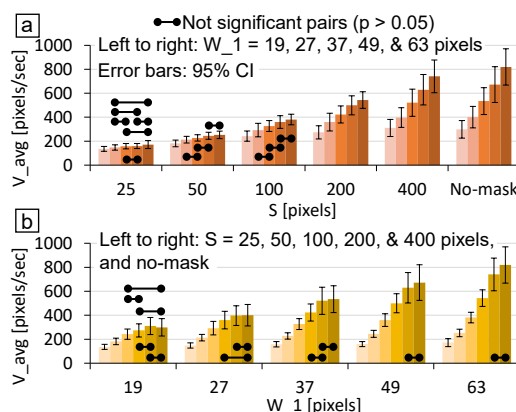

Figure 7: Interaction of $S \times W_1$ on $V_{avg}$. Not significantly different pairs are annotated, and for the other pairs $p < 0.05$.

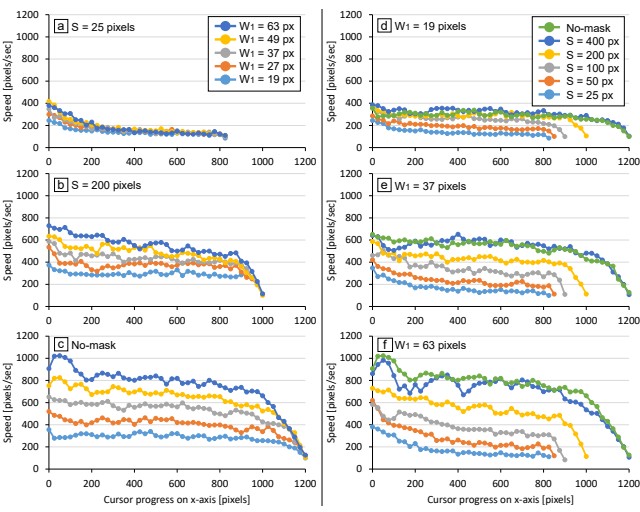

Figure 8: Speed profiles in the measurement area for $A = 800$. (a–c) Speeds for five $W_1$ values for a given $S$. (d–f) Speeds for six $S$ values for a given $W_1$.

We also confirmed that the steering law in the form $V_{avg} = a + bW_1$ (Equation 4) fit reasonably well for each $S$ as shown in Fig. 10. This indicates that the steering law holds when a fixed $S$ is used. In addition, the slopes decreased as $S$ decreased; this means that a small $S$ prohibits a wider $W_1$s from increasing $V_{avg}$.

Fig. 11 shows that the power law ($V_{avg} = a + b\sqrt{S}$, Equation 14 with an intercept) held for large $W_1$ values because the speed was mainly limited by $S$ rather than $W_1$ in this case. However, when $W_1$ was small (19 or 27 pixels), the model fits were degraded to $R^2 = 0.91$. This is because, as statistically shown in Fig. 7b, for small values of $W_1$, the speed was already saturated by $W_1$. This result supports the necessity of accounting for the interaction effect between $S$ and $W_1$ on the speed.

When we used a single regression line of the steering law for $N = 25$ data points ($V_{avg} = a + bW_1$ for $5_S \times 5_{W_1}$ without the no-mask condition), the fitness was poor: $R^2 = 0.180$. This is because $S$ significantly changed $V_{avg}$. Because we have theoretically and empirically shown that $W_1$ and $S$ limit the speed, we would like to integrate both factors.

Table 1: Model fitting results for predicting $V_{avg}$ for $N = 25$ data points ($5_S \times 5_{W_1}$) with adjusted $R^2$ (higher is better) and $AIC$ (lower is better) values. $a$–$d$ are estimated coefficients with their significance levels (*** $p < 0.001$, ** $p < 0.01$, * $p < 0.05$, and no-asterisk for $p > 0.05$) and 95% CIs [lower, upper].

| Model | $a$ | $b$ | $c$ | $d$ | Adj. $R^2$ | $AIC$ |
|---|---|---|---|---|---|---|
| (#1) $a+bW_1$ | 162 [-2.20, 327] | 4.24* [0.331, 8.15] | | | 0.144 | 327 |
| (#2) $a+bS+cW_1$ | 20.5 [-66.8, 108] | 0.914*** [0.695, 1.13] | 4.24*** [2.33, 6.16] | | 0.796 | 292 |
| (#3) $a+bS+cW_1+dW_1S$ | 167*** [87.2, 248] | -0.0342 [-0.423, 0.355] | 0.473 [-1.44, 2.38] | 0.0243*** [0.0151, 0.0336] | 0.912 | 272 |
| (#4) $a+bW_1S$ | 182*** [155, 208] | 0.0241*** [0.0211, 0.0272] | | | 0.916 | 269 |
| (#5) $a+bW_1+cW_1S$ | 162*** [110, 214] | 0.590 [-0.746, 1.93] | 0.0236*** [0.0202, 0.0269] | | 0.916 | 270 |
| (#6) $a+b\sqrt{S}+cW_1$ | -111* [-197, -24.6] | 24.3*** [19.6, 29.0] | 4.24*** [2.63, 5.86] | | 0.855 | 283 |
| (#7) $a+b\sqrt{S}+cW_1+dW_1\sqrt{S}$ | 162*** [95.3, 229] | 0.00463 [-5.36, 5.37] | -2.76** [-4.35, -1.17] | 0.622*** [0.495, 0.750] | 0.974 | 241 |
| (#8) $a+bW_1\sqrt{S}$ | 95.5*** [62.8, 128] | 0.529*** [0.467, 0.592] | | | 0.927 | 265 |
| (#9) $a+bW_1+cW_1\sqrt{S}$ | 162*** [134, 190] | -2.76*** [-3.60, -1.91] | 0.623*** [0.576, 0.669] | | 0.975 | 239 |

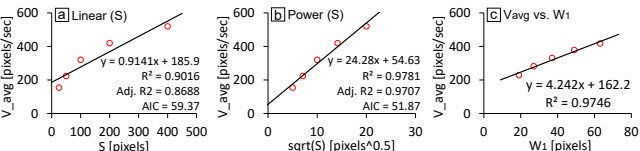

Figure 9: Model fitting results on $V_{avg}$ with (a) $S$, (b) $\sqrt{S}$, and (c) $W_1$.

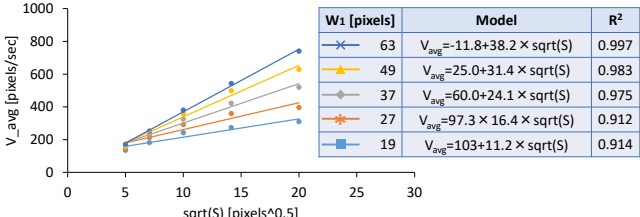

Figure 11: Model fitting results of $V_{avg} = a + b\sqrt{S}$ for each $W_1$.

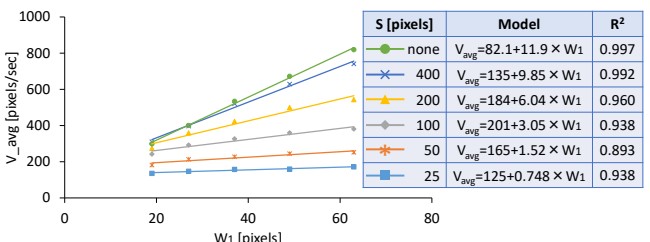

Figure 10: Model fitting results of $V_{avg} = a + bW_1$ for each $S$.

## 4.6 Model Fitness Comparison

To statistically determine the best model, we compared the adjusted $R^2$ and Akaike information criterion ($AIC$) [5]. The $AIC$ balances the number of regression coefficients and the fit to identify the comparatively best model. A model (a) with a lower $AIC$ value is a better one, (b) a model with $AIC \leq (AIC_{minimum} + 2)$ may be as good as that with the minimum $AIC$, and (c) with $AIC \geq (AIC_{minimum} + 10)$ is safely rejected [9].

Table 1 lists the results of model fitting. Model #1 is the baseline model of the steering law. Models #2 to #5 add a new factor ($S$) in a linear function. Because we found a significant main effect of $S$ on $V_{avg}$, Model #2, which adds $S$ to the baseline, improved the fit compared with Model #1. Model #3 adds the interaction factor of $S \times W_1$. Because this term was significant, this model shows a better fit than Model #2.

Models #2 and #3 are simply derived following the statistical analysis method. Note that in Model #3, the main effects of $S$ and $W_1$ were not significant ($p > 0.05$). This means that the effects of $S$ and $W_1$ on increasing $V_{avg}$ can be captured by the interaction factor. Therefore, we also tested the fitness of Model #4. Model #5 was tested for the sake of completeness and consistency with the power model (described below).

Models #6 to #9 are power functions based on Equation 14 with an intercept. Models #6 and #7 are derived similarly to the linear ones. In Model #7, $\sqrt{S}$ was not a significant contributor ($p = 0.999$); Model #9 tests the fit after eliminating this term. For consistent comparison with the linear models, we also tested an interaction-

factor-only model (#8). Model #5 was also tested as a comparison with #9.

Models #7 and #9 are the best-fit models according to their $AIC$ values; the difference between their AIC values was 1.997 ($<2$). Furthermore, the difference in their adjusted $R^2$ values was less than 1%. If the prediction accuracy is not significantly different, a model with fewer free parameters has better utility, and thus, we recommend using Model #9 to predict $V_{avg}$.

By applying Model #9 to Equation 5 (i.e., $MT = a + b[A/V_{avg}]$), we can also predict $MT$ for the measurement area as follows.

$$
\begin{aligned}
MT &= a + b\frac{A}{c + dW_1 + eW_1\sqrt{S}} \\
&= a + \frac{b}{e} \times \frac{A}{c/e + W_1(d/e + \sqrt{S})} \\
&= a + b'\frac{A}{c' + W_1(d' + \sqrt{S})}
\end{aligned}
\tag{15}
$$

For $N = 75$ data points ($= 5_S \times 5_{W_1} \times 3_A$), the baseline steering law model (Equation 6: $MT = a + b[A/W_1]$) showed a poor fit (Fig. 12a). Note that because we discuss the performance in the measurement area, the second-path steering difficulty is irrelevant to the $MT$ prediction presented here; Equation 10, $MT = a + b(A_1/W_1) + cID + d(A_2/W_2)$, is not used. Our model, in which the new steering difficulty is $A/(c' + W_1[d' + \sqrt{S}])$, is able to predict $MT$ more accurately (Fig. 12b).

## 5 Discussion

### 5.1 Findings and Result Validity

In two previous studies on peephole pointing, the error rate was smallest for the smallest $S$ and greatest for the largest $S$ [10, 21]. It was assumed that the participants were overly careful with small $S$ values and overly relaxed with large $S$ values when selecting the target. In contrast, we observed that the error rates tended to be higher for smaller $S$ values. This inconsistency could be because

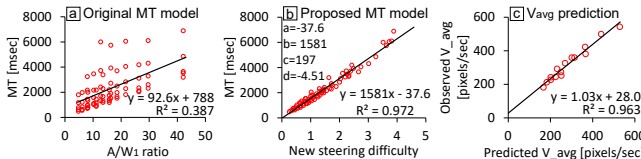

Figure 12: (a) Model fitting for $MT$ using the baseline formulation (Equation 6, $MT = a + b(A/W_1)$). (b) Model fitting for $MT$ using our proposed formulation (Equation 15). (c) Predicting $MT$ as mentioned in the Discussion.

of differing error definitions. In the previous two studies, conventional pointing misses were considered errors, and thus overshooting the target while aiming was permitted, whereas in our study, overshooting at the end of the first path segment was not allowed (i.e., cornering misses were considered errors).

Kaufmann and Ahlström also showed that the movement speed tended to decrease as $S$ increased both with and without prior knowledge ($PK$) of the target position [21]. They explained this in this way: "With small peepholes, participants were eager to uncover the target location by scanning the workspace as quickly as possible; accepting that they would overshoot." Hence, prior to our work, in the HCI field, it has been thought that for peephole interactions the speed decreases as $S$ increases. In our experiment, users could not perform such "quick scanning" because they had to safely turn at the corner.

In previous studies on peephole pointing [10, 21], only the targets were drawn on the workspace, and thus, participants could easily recognize targets via quick scanning. In contrast, in peephole steering such as map navigation [18], quick scanning cannot be performed; if users lose sight of the current road from the peephole window, they have to find the previous road from among several roads and then return to the navigation task. Therefore, we present a new finding on peephole interactions: a larger $S$ increases the speed in overshoot-prohibited conditions [our results] but decreases the speed in overshoot-permitted conditions [10, 21]. This finding contributes to better understanding of users' strategies in the peephole situation.

## 5.2 Other Experimental Design Choices

Our research question regarded the effect of viewable forward distance on the path-steering speed when users have to react to a corner. An alternative is to use a dead end: users must steer through a path and then stop in front of a wall without overshooting. This is called a *targeted-steering task* [13, 22], in which the stopping motion is modeled by Fitts' law. We thus assume that the appropriate model for this task will be similar to our proposed models, but this requires further experiments.

Using only the right-side mask is another possibility for the experiment. We included the left-side mask for consistency with previous works on peephole pointing. Regarding a lasso selection task using a direct input pen tablet (Fig. 1a), the width of the forward mask $W_{mask}$ would affect the speed because the user's hand would occlude the forward path, but beyond the hand, the path would be visible.

While more experimental designs are possible and the resultant speed would change, our experimental data were internally valid. Thus, the fact that "other experimental designs are possible" does not undermine the validity of our models. If user performance under new conditions were to yield different conclusions, that would provide further contributions to the field.

## 5.3 Implications for HCI-related Tasks

Based on our findings, for mouse steering tasks, the speed and $MT$ should change with $S$, but a related work on map navigation with a radar view showed no clear changes in $MT$ [18]. Currently, we have no answer as to whether this inconsistency comes from the fact

that they used a miniature view to show the entire map and/or used magnification or from inaccuracies or blind spots in our own models. The interaction between $S$ and magnification levels on $V_{avg}$ is also unclear and thus should be investigated. As demonstrated in this discussion, our work motivates us to rethink the validity of existing work and opens up new topics to be studied.

Our models could be beneficial in reducing the efforts made to measure users' operation speed for given screen sizes. Once test users operate a map application as in Gutwin and Skopik's study [18] with several screen sizes, the resultant $V_{avg}$ values can be recorded, and we can then predict the $V_{avg}$ for other screen sizes. For example, when we tested only $S = 25$ and 400 pixels ($N = 2_S \times 5_{W_1} = 10$ data points), Model #9 yielded $a = 130$, $b = -2.47$, and $c = 0.622$ with $R^2 = 0.998$. Using these constants, we can predict $V_{avg}$ for $S = 50$, 100, and 200 pixels ($N = 15$), with $R^2 > 0.96$ for predicted vs. observed $V_{avg}$ values (Fig. 12c). Hence, depending on the new screen sizes, the $V_{avg}$ at which test users can perform can be estimated accurately. Importantly, as we showed that the relationship between $S$ and $V_{avg}$ was not linear and that the interaction of $S \times W_1$ was significant, it is difficult to accurately predict $V_{avg}$ for a given $S$ and $W_1$ without our proposed model.

## 5.4 Limitations and Future Work

Our results and discussion are somewhat limited due to the task conditions used in the study, e.g., we did not test circular paths [2] or different curvatures [38]. Also, $A$ ranged from 300 to 800 pixels, but if $A$ is too short or $W_1$ is too wide, the task finishes before the speed reaches its potential maximum value [32, 34]. We limited these values to not be extremely short or wide in order to observe the effects of $S$. Investigating valid ranges of $S$, $W_1$, and $A$ at which our models hold is to be included in our future work.

The width of the second path segment $W_2$ was fixed at 19 pixels and thus was not dealt with as an independent variable. In the derivation of Equation 14 ($V_{avg} = b'\sqrt{S \times W_2} = b''\sqrt{S}$), $V_{avg}$ was originally assumed to increase with $W_2$. This may be true: if users know that $W_2$ is wide, such as 200 pixels, the necessity for quick deceleration would decrease. However, this depends on whether users have prior knowledge $PK$ of $W_2$. If users do not know $W_2$, they have to begin to decelerate as soon as part of the end area is revealed in preparation for a narrow $W_2$. Hence, we have to account for human online response skills, i.e., immediate hand-movement correction in response to a given visual stimulus [24, 40].

$PK$ of the position or timing of when a corner appears would also affect the speed. We tested only a no-$PK$ condition regarding the corner position, which corresponds to conditions in which users do not know the layout of objects in lassoing tasks. In contrast, if users know the layout, control is possible at much higher speeds while avoiding unintended selection. A kind of *medium-PK* is also possible in our experiment. That is, although the participants did not know $A$ beforehand, if the second path segment did not appear on the left half of the screen, the participants realized that they needed to decelerate because the corner must have been in the remaining space. Employing a complete no-$PK$ condition to evaluate peephole pointing and steering would therefore be difficult for desktop environments, although this technical limitation has not been explicitly mentioned in the literature [10, 16, 21, 30].

## 6 CONCLUSION

We presented an experiment to investigate the effects of viewable forward distance $S$ on path-steering speeds. In the path-steering tasks with cornering at an uncertain time, the relationship between $S$ and the speed followed a power law (square root), and the interaction between path width $W_1$ and $S$ was accounted for to accurately predict the speed. The best-fit model showed an adjusted $R^2 = 0.975$ with only one additional constant added to the baseline steering law, which also yielded an accurate model for task

completion times. Interestingly, opposite conclusions were derived depending on the task requirements; a shorter $S$ increased the speed in peephole pointing [10, 21] but decreased it in our path-steering experiment. Although few studies have focused on the effects of $S$ on user performance, the importance of this topic will increase with the growth of devices with limited view areas, such as smartphones and tablets, and thus, we hope this topic is revisited by many more researchers in the future.

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
