# OpenReview forum: "Peephole Steering: Speed Limitation Models for Steering Performance in Restricted View Sizes"
_graphicsinterface.org/Graphics_Interface/2020/Conference — GI 2020_

### Official Review · AnonReviewer2 · 2020-04-17
**Sound study executed well with good analysis, need better sample use cases**

**Rating:** 6
**Confidence:** 4

**Review:**

This paper reports empirical results of a study investigating how viewable forward distance affects steering error and speed in a steering tasks where a corner turn is present. The paper is articulated well and the study and analysis are sound. Overall I think it makes a small but original and useful contribution in terms of modeling performances for steering tasks. I do however think the introduction is a bit over-promising without adequate backings. For example, it talks about possible utility in lasso selection and VR environments, but the study in the paper used mouse only and the steering task was a 2D movement with just one turn. Nevertheless if the authors could elaborate more on its applications, or provide a more reasonable scope, I'm inclined in accepting this paper.

Pros:
-The explanation of models and the study is well-written. The inclusion of apparatus choice and system setting is useful for replication.
-The authors did a good job in discussing other experimental design choices, which I think warrant further investigation like left-hand masks and for people who use a different hand.
-It is good for the authors to state and discuss inconsistent results with prior work.

Cons:
-There are several parts in the paper that require clarifications:
--In related work the authors claimed that "If Bateman et al. had tested longer S values like 100 m, S might have significantly affected the results", but offered no explanation. Please back this claim.
--It seems to me that the authors used two different models to estimate MT for the first and second path segments. If that's the case, why? Could one use the same model (eq. 10) for the second path as well? Please elaborate the rationale behind, or make it clear which model was used.
--Choice of some experimental parameters were not explained well. Base on what did the authors "set a narrow W2 requiring careful movements to safely turn at the corner"? What makes it narrow enough? Why did the corner always turn downwards in the study?
--Some data were left unexplained. Why the slightly out-of-order jump of error in S for 4th and 6th widths? I'd have expected the wider S gets the less error one makes so the trend will be monotonous. It is also strange to have some of the results (error rate analysis) attached as supplementary materials.
--In the discussion of limitation the authors said they tested only no-PK conditions, but during the model development they said the participants had the previous knowledge of W2 being fixed. It appears that here the authors were refereing to A instead W2. I'd suggest clarifying what was known and what was not known by the participants.

-Lastly, I think the driving simulators and racing games examples provided in the future work aren't appropriate. This is because braking (pressing/releasing a button) is different from slowing down wrist movements, and the way the player avoids running to the sides is different from the pointer hitting the walls. I think lassoing would be a better example. There is nothing wrong with limiting the scope to what the study actually reveals.

Minor points:
-missing example in a few places such as abstract and related work (do a search for "(e.g.)").
-it would be easier for readers to understand what "viewable forward distance" is if Figure 1a is annotated.
-Equation 14 in the sentence "Equation 14 can be simplified further" should be Equation 13?
-I'm not certain what the authors mean by "human online response skills" when discussing future work. Please explain.

---

### Official Review · AnonReviewer1 · 2020-04-20
**Small and specific problem, but well investigated.**

**Rating:** 6
**Confidence:** 3

**Review:**

This paper proposes a steering speed model that takes path visibility into account. It provides a theoretical discussion to justify specific aspects of the model, and reports a controlled study in which various combinations of parameters were tested as an extension of the steering law. The results can inform the design of steering tasks in which parts of the path is occluded, and overshoot and clutching are forbidden.

Overall I am in favor of accepting this paper, which I try to explain below

# Strengths of the paper
The paper is well written and addresses a specific but interesting aspect of steering models. The "hand occlusion when tracing" example that is shown on figure 1 is a nice example use-case.

The related work section clearly describe existing steering law models, as well as discussing research on other yet related research area (and typically peephole pointing or scrolling models), making it interesting and rather complete regarding the scope of the problem explored in the paper.

The theoretical discussion of models that can play a role in speed limitations for steering performances with restricted view is very clear, easy to follow, and it is easy to understand why the proposed model might be adapted.

The experiment reported in the paper, while testing a limited number of parameters and conditions, is well described and sounds relevant.

Finally, the discussion section provides a short but neat discussion of the results, and I especially appreciate that the authors discuss few of the their results who were inconsistent with the literature.

The final model, while useful for a very niche task (see below), is clearly good.

# Weaknesses and limitations
While sound and interesting, the proposed model remains focused on a very specific use case that is, in my humble opinion, not so frequent in everyday interaction with computing systems (or at least, not in the way it was tested in this submission). As mentioned above, I especially liked the "hand occlusion when tracing" use case example, but I would not necessarily consider all other examples (namely, 3D drawing or steering with head-mounted displays or racing games/driving simulators) provided as relevant in this context. Typically, in the context of HMDs, users often have distinct and integrated control of viewport and pointing device through two dedicated devices. Regarding racing/driving apps, they tend to rely on rate control rather than position control. In addition, when reading the introduction of the paper I was always thinking of when I import a photo in illustrator, zoom on it, and then draw contours on top of it using my mouse which "autoscrolls" the viewport so I can move all around the figure I am contouring. All these scenarios, while similar, differ from the experimental setup in term of how users control scrolling and cursor position (when relevant). As such, the model may or may not apply. I would recommend to remove these examples from the paper and would focus the examples on lassoing and hand occlusion when tracing uses cases, which would avoid potential overclaims.

Another weakness in my opinion lies in how experimental results are reported. While clear and easy to follow, authors on several occasions provide null hypothesis testing result, reporting significant differences, but without providing post-hoc tests (e.g ErrSteer results). Such post-hoc tests, while not the core of the paper, would still be interesting. In its current form, the results are not complete and significant differences between conditions must be inferred from figures.


That being said, while of limited contribution, the contribution remains significant and well explained which is why I remain in favor of accepting this paper.

---

### Official Review · AnonReviewer3 · 2020-04-21
**Solid modeling paper with sound empirical evaluation.**

**Rating:** 8
**Confidence:** 3

**Review:**

This paper proposes a statistical model for steering control.

Understanding user behavior in a restricted visibility setting is definitely a topic of interest in the HCI community. While I am excited about this paper, I would like to raise a few points for the authors to further clarify to strengthen the presentation and the readability of the paper. I hope the authors find them useful.

Theoretically, this is an interesting cognition scenario. While we are known to employ intermittent control, we cannot “plan” as much as we like when “view” is limited. However, in practice, even with more devices with different screen estates make the scenario more common than before, it would help the readers if the authors provided more examples of when occluded steering scenarios occur.

The experiments are solid. Although somewhat limited in task setup - like fixing segment widths. However, it would help the readers if some of the choices were elaborated more, for example, what makes a path "narrow"? It would also help the readers if the results were unpacked more and contextualized. For example, I am still having difficulty understanding what it means to observe the interaction of S × W1 being statistically significant.

Lastly, I would like the authors to draw more implications for designers. For example, what do the results of this study provide for building computational models of steering with prediction capabilities? What are the takeaways for building a simulator?

The paper is well written and well structured.

---

### Meta-Review · Area_Chair1 · 2020-04-22

**Recommendation:** Accept
**Confidence:** 4

**Metareview:**

Overall, reviewers were on agreement about the  quality of the paper and the fact that it is well written and structured, investigating an interesting topic and describing a solid experiment. Reviewers (R1,R2) also praised that while the contribution is small, it remains useful to the HCI community.

That being said, reviewers also mentioned several weaknesses in the current submission that the authors should address, as detailed below.

# Use cases
Reviewers had mixed feelings regarding the use-case examples presented in the papers. On one hand, R1 and R2 found that some examples should be removed and that the paper should be more focused on the use cases on which it is clear that the proposed model applies. On the other hand, R3 would like the authors to provide more example. I would recommend the authors to go with the former and to focus the paper on a very specific, yet existing, use case. As R2 says, "There is nothing wrong with limiting the scope to what the study actually reveals.".

# Clarify decisions
R2 points that authors should clarify why two different models are used to estimate MT for the different path of the segments. Authors should clarify why this specific use requires these two models (due to the cornering).

Also, R2 would like the authors to better justify some experimental design decisions. Typically, why was W2 sufficiently narrow for this task.

# Study results
Reviewers also expressed some concerns regarding how results are reported.

R1 and R2 raised that post-hoc tests are not mentioned in the paper, and that pairwise significant differences are only present on some figures (or not present at all, only in supplementary materials). The authors should provide all these results for the sake of clarity and completeness.

R3 mentions that it would help the readers if the results were unpacked more and contextualized, providing interaction effect as an example. Personally, I believe that it is not necessary for clarity, but agree that it provide a better reading experience

Finally, R2 regrets that some data were left unexplained (typically the slightly out-of-order jump of error in S), and authors should provide more explanations regarding this discussion

#Discussion section
Given above modifications, the authors should reshape the discussion section to better insist on the limitations of their study. More precisely, reminding what was known or not by the participants, and to which extent different variables were tested or not (such as the fact that only one W2 values was tested).
Also, the authors should clearly acknowledge that future work is needed to confirm that the model would work in a similar way with other use cases, such as racing games or HMDs interaction.

# Other recommendations
- Complete or remove claim regarding "Bateman et al." work.
- Fix typo with (e.g.) in the abstract
- Clarify sentence "Equation 14 can be simplified further" as pointed out by R2
- Explain what "human online response skills" refers to for readers who are not familiar with this concept.
- Update figure 1 to remove examples the study does not apply to, and highlight "viewable forward distance" on it.

---

### Decision · Program_Chairs · 2020-04-25

Accept